

# Above and belowground phenotypic response to exogenous auxin across *Arabidopsis thaliana* mutants and natural accessions varies from seedling to reproductive maturity

Patrick Sydow[1,2] and Courtney J. Murren[1]

[1] Department of Biology, College of Charleston, Charleston, SC, United States
[2] Department of Plant Science, The Pennsylvania State University, University Park, PA, United States

## ABSTRACT

**Background:** Plant hormones influence phenology, development, and function of above and belowground plant structures. In seedlings, auxin influences the initiation and development of lateral roots and root systems. How auxin-related genes influence root initiation at early life stages has been investigated from numerous perspectives. There is a gap in our understanding of how these genes influence root size through the life cycle and in mature plants. Across development, the influence of a particular gene on plant phenotypes is partly regulated by the addition of a poly-A tail to mRNA transcripts *via* alternative polyadenylation (APA). Auxin related genes have documented variation in APA, with auxin itself contributing to APA site switches. Studies of the influence of exogenous auxin on natural plant accessions and mutants of auxin pathway gene families exhibiting variation in APA are required for a more complete understanding of genotype by development by hormone interactions in whole plant and fitness traits.

**Methods:** We studied *Arabidopsis thaliana* homozygous mutant lines with inserts in auxin-related genes previously identified to exhibit variation in number of APA sites. Our growth chamber experiment included wildtype Col-0 controls, mutant lines, and natural accession phytometers. We applied exogenous auxin through the life cycle. We quantified belowground and aboveground phenotypes in 14 day old, 21 day old seedlings and plants at reproductive maturity. We contrasted root, rosette and flowering phenotypes across wildtype, auxin mutant, and natural accession lines, APA groups, hormone treatments, and life stages using general linear models.

**Results:** The root systems and rosettes of mutant lines in auxin related genes varied in response to auxin applications across life stages and varied between genotypes within life stages. In seedlings, exposure to auxin decreased size, but increased lateral root density, whereas at reproductive maturity, plants displayed greater aboveground mass and total root length. These differences may in part be due to a shift which delayed the reproductive stage when plants were treated with auxin. Root traits of auxin related mutants depended on the number of APA sites of mutant genes and the plant's developmental stage. Mutants with inserts in genes with many APA sites exhibited lower early seedling belowground biomass than those with few APA sites

Corresponding author
Courtney J. Murren,
murrenc@cofc.edu

but only when exposed to exogenous auxin. As we observed different responses to exogenous auxin across the life cycle, we advocate for further studies of belowground traits and hormones at reproductive maturity. Studying phenotypic variation of genotypes across life stages and hormone environments will uncover additional shared patterns across traits, assisting efforts to potentially reach breeding targets and enhance our understanding of variation of genotypes in natural systems.

## INTRODUCTION

The structure of root systems and their subsequent functions vary across plants and environments (*Weemstra et al., 2021*). In particular, the architecture of root systems contributes to a plant's capacity to anchor in place and acquire nutrients and water (*Smith & De Smet, 2012*). Furthermore, lateral roots allow for branching root growth away from the primary root which contributes to root positioning in root systems (*Jones & Ljung, 2012*). Changes in the number of lateral roots produced during development are mediated by auxin *via* a suite of auxin related genes (*Du & Scheres, 2018*). Through signal transduction, the transport and biosynthesis of auxin contribute to the signaling and control of cell division, cell growth, and differentiation across seedling root systems (*Roychoudhry & Kepinski, 2022*). Additional regulation by auxin on root systems occurs through the post-transcriptional modification of auxin and signal transduction transcripts through the alternative polyadenylation of the transcript (*Hong et al., 2018*). While the direct investigation of the role of auxin and auxin related genes has been studied extensively at the early seedling stage (*Casimiro et al., 2001*; *Shukla et al., 2019*), auxin's influence on lateral roots and root size at later stages of development is less documented offering opportunities for further investigation in older seedlings and at reproductive maturity. Alternative polyadenylation, a form of post-transcriptional modification, can regulate various aspects of plant development (*Hunt, 2022*), and as a consequence may shift trait expression from seedlings to maturity. Alternative polyadenylation is widespread across almost 100 auxin and auxin associated genes (*Hong et al., 2018*) and is hypothesized to alter the function of auxin related genes across the life cycle of a plant. Thus, specific auxin related genes may have varying control of root growth across plant development. Here we address three gaps in the literature to provide additional evolutionary and ecological context to studies of auxin related genes in seedlings. Specifically, we investigate: (1) developmental variation in belowground and aboveground phenotypes of auxin related mutants in treatments with exogenous auxin applications, (2) if mutant gene APA site number explains variation in phenotypes across development, and (3) how perturbations in auxin related genes in combination with additional auxin affect measures of plant fitness to examine the potential for evolutionary-ecological mechanisms shaping variation.

Auxin's effect on root development has been studied extensively in *A. thaliana* seedlings younger than 14 days (*Casimiro et al., 2001*; *Krouk et al., 2010*; *Marin et al., 2010*; *Xuan*

*et al., 2016*; *van Gelderen et al., 2018*; *Shukla et al., 2019*; *Yang et al., 2020*), showing that auxin induces lateral root development in seedlings 8 to 12 days after germination. Indications are that auxin continues to have a role in root traits such as lateral roots as the plant grows, although data quantifying these patterns at later developmental stages are scarce (*Gâteblé & Pastor, 2006*; *Osterc & Štampar, 2011*; *Rinaldi et al., 2012*; *Bao et al., 2014*). While some genes may have a promote lateral root development at the early seedling stage (*e.g.*, auxin transporters in *Casimiro et al., 2001*), the same gene may ultimately lead to diminished fruit production, a measure of fitness, due to potential negative effects on root structure and function in later development.

The most prominently studied auxin function in seedling root systems is in the initiation, emergence, and growth of lateral roots (*Lavenus et al., 2013*; *Du & Scheres, 2018*). Genes within the auxin biosynthesis pathway mediate the development of the root meristem (*Olatunji, Geelen & Verstraeten, 2017*). Whether through changes in transport (*Krouk et al., 2010*) or hormone regulated genetic mechanisms (*Pérez-Torres et al., 2008*), auxin influences roots structure in large part through lateral root production. Auxin biosynthesis, signaling, and transport respond to abiotic and biotic factors and interact with other plant hormones through development (*Kazan, 2013*). Thus, the ways in which specific auxin related genes alter lateral roots and root structure at different development timepoints may ultimately enhance reproductive fitness. Early seedling assays are only part of the story. The continued growth of a greater density of lateral roots observed at the seedling stage may lead to larger root systems contributing to greater resource acquisition in later development, which can result in enhanced fitness.

Expansion of genetic variation occurs through mutations and post transcriptional modification. Post-transcriptional modification can occur through alternative polyadenylation (*Zhao, Roundtree & He, 2017*). Similar to alternative splicing of RNA, alternative polyadenylation (APA) allows a single gene to produce multiple mRNA transcripts which can have different functions altering the expression of genes (*e.g.*, *Ye et al., 2019*). Polyadenylation of the 3′ end of eukaryotic mRNA involves the addition of a poly(A) tail to the pre-mRNA within the nucleus which affects mRNA metabolism by influencing mRNA stability, transport, and translation efficiency (*Hunt, 2022*).
In Arabidopsis, genome-wide analyses have revealed that APA occurs extensively and throughout the genome. Estimates suggest that up to 70% of Arabidopsis genes contain alternative sites which could produce various RNA transcripts and proteins through this mechanism (*Wu et al., 2011*), which could contribute to enhanced variation in the effects of particular genes through the lifecycle of the plant and across plant traits. Auxin in particular is observed to alter APA by changing poly(A) site usage in genes that affect lateral root growth.

While IAA (indole-3-acetic acid) is the most common auxin in plants with IAA and other auxins interact with other proteins to promote the transcription of auxin related genes (*e.g.*, TIR1/AFB auxin co-receptors). Regulation of auxin genes occurs through the degradation of Aux/IAA proteins which work with DNA bound ARF (auxin response factor) transcription factors that repress and activate the transcription of auxin related genes (*Cancé et al., 2022*), suggesting that distinct gene families with dynamic functions

may play a role in phenotypic variation in roots across development in response to auxin (*Du & Scheres, 2018*). The RNA transcripts of both ARF and IAA genes (*ARF7*, *ARF19* and *IAA14*) are altered *via* poly(A) site switches in response to auxin (*Hong et al., 2018*). Thus, we hypothesize that APA may be an important contributor to differences in auxin mediated root growth and such variation in root traits may be altered at distinct developmental stages. Further we hypothesize that greater number of poly(A) sites contribute to differential gene function across the lifecycle of a plant by allowing a diverse suite of potential RNA transcripts when compared to genes that do not exhibit polyadenylation. From a phenotypic perspective, we hypothesize genes with fewer APA sites make consistent contributions to organ development across the lifecycle of a plant, thus mutations in these genes having a greater effect on whole plant phenotypes and fitness than those with many APA sites.

Here, we evaluate a suite of *Arabidopsis thaliana* insert mutants across a diverse set of auxin related genes to evaluate the contribution of auxin, specific auxin pathways, and alternative polyadenylation to variation in plant fitness and whole plant phenotypes across development. Specifically, we used an available library of *A. thaliana* insertion mutants (*O'Malley & Ecker, 2010*; *Rutter et al., 2017*, *2019*) and a suite of natural population and applied auxin exogenously to roots throughout development to examine the following questions.

(1) Do mutant lines of key auxin pathway genes in the ARF and IAA families vary in their response to exogenous treatments of auxin in the form of IAA? We predict that lines with insertion mutations in auxin pathway genes will vary in their response to exogenous auxin treatments showing differences in belowground and aboveground phenotypes paired with differences in plant fitness quantified as fruit production.

(2) Do differences in root traits of mutant lines and natural accessions change across developmental stages? We hypothesized that root traits in all genotypes would differ such that root traits (root length and biomass) would be greater in later developmental stages and vary significantly across mutant lines in rank order across developmental stages as we anticipate gene and gene family specific influences at particular stages.

(3) Do the root phenotypic responses of mutant lines of auxin genes correspond with the number of alternative polyadenylation sites? Is this pattern the same in seedlings and reproductive adult plants? We predicted that genes with fewer poly(A) sites would function similarly across plant development assessed *via* root trait phenotypes when compared to mutants with inserts in genes with many poly(A) sites that may confer functional variation across the lifecycle of the plant. Specifically, we expect mutants of genes with fewer poly(A) sites to exhibit greater changes in plant phenotypes and diminished fitness with decreased fruit number, less overall root length, and smaller rosettes in comparison to wildtype control.

(4) Is the phenotypic variation across auxin related gene mutant lines comparable to that produced by natural variation across multiple natural accessions? We predict that natural accessions that exhibit substantial variation across the genome will produce a wider range of phenotypes to auxin treatments than insert mutants which share genetic backgrounds and only differ in a mutation at one locus. Together, examining these

questions will allow us to further refine our understanding of the function of auxin related genes across development adding later stage context to early seedling assays and aid in identifying target genes for crop improvement and potential for root variation in natural ecosystems.

## MATERIALS AND METHODS

### Study species and line selection

*Arabidopsis thaliana* is an herbaceous annual species native to diverse environments in Europe, West Asia, and Africa (*Hoffmann, 2002*; *Fulgione & Hancock, 2018*). Populations differ in their root traits and soil nutrient characteristics which can be selective agents for root architectures, expose variation among mutant lines (*Murren et al., 2020a*) and confer fitness advantages in the wild (*Murren et al., 2020b*, *2022*). As a winter annual, germination typically occurs within 5 days following cold stratification and flowering typically occurs 27 days following germination (*Rutter et al., 2017*, *2019*). Six spring-flowering winter annual natural accessions were randomly selected to serve as phytometers (see *Rutter et al., 2019* for additional details) for the mutant lines.

We chose mutants from seed stock sourced from the SALK unimutant collection containing lines with T-DNA inserts. To study the variation in the response to auxin of auxin-related genes, we selected those mutant lines that were homozygous for the insert in a single auxin-related gene (*O'Malley & Ecker, 2010*; *Rutter et al., 2019*). We selected SALK lines with inserts in genes involved in auxin transport, synthesis, metabolism, and signal transduction and replicates of representatives of two gene families: Auxin Response Factor (ARF) and Aux/IAA (Auxin/Indole-3-Acetic Acid). We restricted our selection to only those with a single insertion mutation (see *Rutter et al., 2019* for details on screens). From this longer list of mutant lines, we selected lines that fell in to three groups: those genes with 0 APA sites, genes with up to two APA sites and genes with >2 APA sites using data in *Hong et al. (2018)*. This resulted in replicate lines in each of the three categories (>2 sites $n = 4$, 1 + 2 sites $n = 6$, 0 sites $n = 5$) that we used test for differences in effects on plant phenotypes across genes that vary in numbers of APA sites. Supplemental S1 includes information on genes, their polyadenylation, and respective SALK lines along with annotations of mutant loci with APA sites and T-DNA insert locations (Table S1; visualizations of insert and APA sites in Fig. S1).

All plants were grown within Percival growth chambers (Percival-Scientific, Ames, IA, USA) in standard growing conditions: 22 °C with 16-h photoperiod (following methods in *Rutter et al., 2019*). Following vernalization at 5 °C for 1 week on wet filter paper, seeds were sown onto a substrate of rinsed sand (Play Sand; Quikrete, Atlanta, GA, USA) and soil conditioner (Soil Perfector; Espoma, Millville, NJ, US) in a 2:1 ratio (see as descriptions in *Cousins & Murren, 2017*) for control and auxin treatments. Each well was pretreated with 2 mL of 100% Hoagland's nutrient solution (PhytoTech Labs, Shawnee Mission, KS, USA) in deionized water before sowing seeds.

## Auxin treatments

We replicated previously used concentrations of exogenously applied auxin which elicited a growth response in seedlings on agar (*Wilmoth et al., 2005*), and significant phenotypic responses with this level were confirmed in our pilot experiments in our potting media for aboveground and belowground traits. Our goal was to implement one auxin addition treatment to test mutant and developmental responses to exogenous auxin. Auxin treatments were applied directly to the substrate upon germination using 1 mL of a $1.0 \times 10^{-4}$ M IAA solution in which IAA (I2886; Sigma-Aldrich, St Louis, MO, USA) dissolved in deionized water and treated weekly using 2 mL of a $1.0 \times 10^{-4}$ M IAA and 100% Hoagland's minimal nutrient solution. Control plants were treated with 2 mL of 100% Hoagland's solution weekly. All plants were watered from below as needed. All lines were also grown in PRO-MIX (Premier Tech Horticulture, Quakertown, PA, USA) as an additional control for seed germination.

## Replication and developmental stages

Developmental stages and root sampling times were determined following a review of existing literature (Supplemental S2). To study early seedling root structures, the 21 genotypes described above were grown for 14 days (early seedling) or 21 days (late seedling), with six replicates per treatment (sand control or IAA) organized in a randomized block design across plug trays (504 plants). To study root structure variation at reproductive maturity, we grew the plants until their most basal siliques dehisced. We grew six replicates per treatment (sand control, IAA, or PRO-MIX control), except for the wildtype Col-0 (CS70000) which we increased replication to 18 plants per treatment. Plants in this experiment were grown in a randomized block design and sown across plug trays in one growth chamber (414 plants).

## Measurements

We measured germination, rosette size, root traits, and biomass. Flowering and fruit traits were measured in the adult plants. Germination was recorded every day until all experimental seeds germinated. When we harvested seedlings, rosette diameter was recorded for each plant as a measure of aboveground size. We also noted if bolting had commenced, the date of onset of bolting was recorded. To harvest the roots, we cut the plant at the hypocotyl just below the rosette and removed the entire sand plug from the tray. Roots were then cleaned using a paintbrush and by rinsing the root with water before being stored in Whirl-Pak sampling bags (Nasco, Madison, WI, US) prior to scanning as in *Cousins & Murren (2017)* and *Murren et al. (2022)*. To make the quantitative root measurements, all roots were placed in a tray of water positioned to minimize overlap prior to scanning using an Epson Expression 10000XL scanner (Seiko Epson, Nagano, Japan). The total root length of each root system was then measured using WinRHIZO software with scaled calibration (Regent Instruments, Ville de Québec, QC, Canada). Following scanning, we dried roots in weigh paper and placed roots along with aboveground portions of plants in a forced-air safety drying oven (VWR, Radnor, PA, US) at 37 °C until a

constant mass was reached. The belowground portion of plants grown in PRO-MIX was not collected as the potting medium could not be fully washed from the roots.

We used Image J (National Institutes of Health, Bethesda, MD, USA) to measure the length of the primary root for 14 and 21 d seedlings. The seedling lateral root density was calculated by dividing the sum of the number of upper, mid, and lower lateral roots by the length of the primary root. The lateral root density of 21 d roots was calculated by dividing the sum of mid and lower lateral root count by two-thirds of the length of the primary root.

For plants grown to maturity, days to bolt, rosette diameter at bolt (defined as when the inflorescence was >5 mm), days to flower, and days to first mature fruit were recorded throughout the experiment. The experiment was terminated after 57 days when >98% of bolted plants reached reproductive maturity.

We also measured architectural and reproductive aboveground phenotypes of plants grown to maturity including the number of branches produced; the number of fruit, flowers, flower buds, and aborted fruit; basal, mid, and upper fruit lengths; and total fruit number per inflorescence, and total inflorescence height following *Rutter et al. (2019)*. Roots of mature plants were scanned for total root length using methods described above. Aboveground biomass and belowground biomass were recorded after we brought the roots and shoots to room temperature in sealed containers containing Drierite (WA Hammond Drierite, Xenia, OH, US). Aboveground biomass measurements for the seedlings and belowground biomass measurements for all plants were measured using a microanalytical balance (Excellence Plus XP6; Mettler Toledo, Columbus, OH, USA) and aboveground biomass of the mature fruiting plants were quantified using a mass balance (OHAUS Explorer E12130; Parsippany, NJ, USA).

## Statistical analyses

To test genotype (mutant line or natural accession), development stage and exogenous auxin treatment effects on phenotypes we used linear and linear mixed models created in R (version 4.0.3; *R Core Team, 2021*). When examining differences between genotypes, treatments, and their interaction within a developmental stage, stage specific traits were analyzed (*e.g.*, phenology and fitness in mature plants and total lateral root density in early seedlings). To examine differences in genotypes across developmental stages within each treatment, we focused on root length, belowground mass, and aboveground mass measured at each of the three developmental stages. Traits were square root or log transformed when necessary to meet assumptions of ANOVA.

Plant response to exogenous auxin—To determine how insert mutant line phenotypes vary in their response to exogenous IAA, linear models were created in R using the stats package (R version 4.0.3; *R Core Team, 2020*). We examined models for rosette diameter, root length, belowground biomass, aboveground biomass at each developmental stage. We examined lateral root density (LR density) separately at the two seedling stages and the timing of phenological changes and fitness traits in plants grown to maturity. In these stage specific models, measured traits were included as response variables and hormone treatment, genotype (including only mutant lines), and treatment by genotype interaction terms were included as fixed effects. We compared the phenotypes of the two gene families:

IAA ($n = 4$) and ARF ($n = 5$) mutants and their response to the auxin addition treatment in comparison to the control.

Phenotypic variation and development—To examine how auxin mutant genotypes and natural accessions vary across developmental stages in phenotypic traits, linear and linear mixed models were created (using the lme4 package version 1.1–27.1 for mixed models; *Bates et al., 2015*). We analyzed traits measured at 14 and 21 d and traits measured at all three developmental stages separately. We normalized trait values by developmental stage and treatment. In linear models, traits measured only at the seedling stage were included as response variables. Developmental stage (defined as 14 days for early seedlings, 21 days for late seedlings), genotype (including mutant lines and natural accessions), and developmental stage by genotype interaction terms were included as fixed effects. We compared LR density of the bottom two thirds of the primary root across seedling stages. Linear mixed models across entire plant development included tray as a random effect.

Variation among APA mutant groups across development—We asked how the number of APA sites of a line with a mutation in the gene influences changes in plant phenotypes at distinct developmental stages. To do so we created linear mixed models using the lme4 package. We normalized traits by developmental stage and auxin treatment. Developmental stage, APA group (mutant lines grouped into three levels: >2, 1–2, 0), and developmental stage by APA group interaction were fixed effects. Tray was included as a random effect.

Variation of natural accession and mutant phenotypes—To examine differences between the trait variation in the phenotypes of natural accession and insert mutants, we used Levene's tests in R using the car package (version 3.0-11; *Fox & Weisberg, 2019*).

Fixed effects in linear and linear mixed models were tested using anova function in the stats package in R (version 4.0.3; *R Core Team, 2020*). Random effects in linear mixed models were tested using the ranova function in the lmerTest package (version 3.1-3; *Kuznetsova, Brockhoff & Christensen, 2017*). Significant differences between pairs of APA groups within models were determined using TukeyHSD-like *post-hoc* tests of linear mixed models using emmeans (version 1.7.2; *Lenth, 2022*). Data manipulation was completed using dpylr (version 1.0.7; *Wickham et al., 2021*) and tidyr (version 1.1.4; *Wickham, 2021*). Figures were produced using ggplot2 (version 3.3.4; *Wickham, 2016*), gridExtra (version 2.3; *Auguie, 2017*), scales (version 1.1.1; *Wickham & Seidel, 2022*), and GIMP (*GIMP, 2022*).

## RESULTS

### Plant response to exogenous auxin

Across traits, we detected a significant influence of exogenous auxin on aboveground and belowground plant phenotypes and significant variation among genotypes. While early seedlings exhibited root systems with significantly shorter length and higher LR density in the auxin treatment, late seedlings when treated with auxin exhibited significantly higher LR density ($F = 27.9$, $p = 5.715e-07$) but no difference in root length (Fig. 1; Tables S2 and S3). We found significantly smaller rosettes in early seedling mutants treated with auxin

($F$ = 22.3, $p$ = 5.5e-06) and decreased biomass aboveground ($F$ = 32.8, $p$ = 6.1e-08) and belowground ($F$ = 4.3, $p$ = 0.041; Tables S3 and S2). Late stage seedling mutants exhibited significantly decreased aboveground biomass when treated with auxin ($F$ = 45.5, $p$ = 4.48e-10) with no change in belowground biomass or rosette diameter (Table S4). Rosette diameter ($F$ = 4.5, $p$ = 5.5e-06), root system length (Fig. 1), and aboveground biomass ($F$ = 3.5, $p$ = 6.5e-05) and belowground biomass ($F$ = 3.7, $p$ = 3.5e-05) differed significantly between genotypes at the early seedling stage. We did not detect any statistically significant interactions between genotype and auxin treatment in the early or late seedling stage (Tables S2 and S4).

We detected phenotypic differences between auxin treatments at early life stages and in vegetative characters, but not in key reproductive fitness characters. Insert mutants at the mature fruiting stage treated with auxin exhibited increased plant vigor with significantly greater measures of rosette diameter at bolting ($F$ = 15.2, $p$ = 0.00015), root system length (Fig. 1), belowground mass ($F$ = 84.5, $p$= 5.6e-16), and aboveground mass ($F$ = 15.6, $p$ = 0.00013; Tables S5 and S10). While inflorescence height ($F$ = 10.3, $p$ = 0.0017) and average fruit length ($F$ = 27, $p$ = 7.3e-07) of all mutants were significantly higher when plants were treated with auxin, key measures of fitness (fruit number and inflorescence branching) did not differ significantly between treatments (Tables S5 and S10). The timing of the transition between life stages was significantly delayed when mutants were treated with auxin with plants expressing later bolting, flowering, and fruit ripening (days to mature) (Fig. 2 and Table S5).

Genotypes differed significantly for a number of aboveground and belowground traits, such that mutant lines varied in their phenotypic response but were trait specific. Genotypes differed significantly in their inflorescence height ($F$ = 1.9, $p$ = 0.035) as well as time to bolting ($F$ = 1.8, $p$ = 0.037), flowering (Fig. 1), and days to fruit maturity ($F$ = 2.1, $p$ = 0.015; Table S5). Most mutants exhibited a later flowering time in the auxin treatment (Fig. 1), but *sur1*, *arf21*, and *arf4* mutant genotypes exhibited an earlier flowering time when treated with auxin (Fig. 1 and Table S5). Contrasting ARF and IAA gene family mutants, we found Auxin Response Factor genes (Fig 3 and Table S1) exhibited delayed flowering in both treatments. In contrast, lines with T-DNA inserts in genes within the IAA family display earlier flowering without the addition of auxin (Fig. 3). No detectable differences were observed among genes categories: auxin synthesis and metabolism, auxin polar transport, and auxin signal transduction (Table S1).

**Phenotypic variation and development**

We found that root traits, when scaled by developmental stage and treatment, displayed significant differences in the rank of genotypes across seedling stages, such that stage and gene influenced which plants had the largest roots (Table S7). In seedlings treated with auxin, a significant developmental stage by genotype interaction was observed for lateral root (LR) density. The *arf14* mutant had decreased LR density in the late stage in comparison to other genotypes including wildtype ($F$ = 2.1, $p$ = 0.0059; Fig. 4).

We examined root length, belowground mass, and aboveground mass across the life cycle of *A. thaliana* (Table S8) and found treatment specific effects in genotype variation

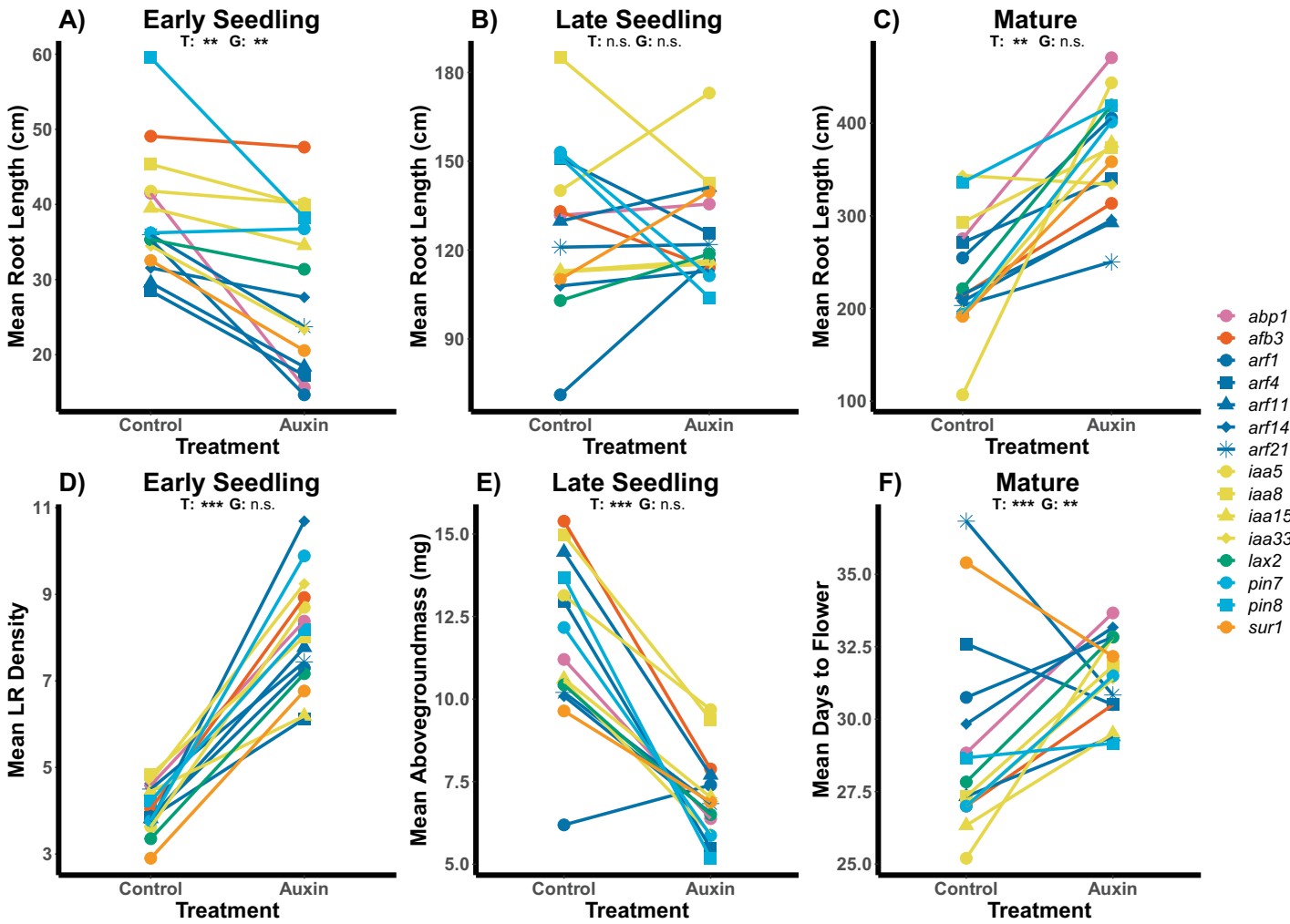

**Figure 1 Effects of auxin treatments on mutant root and shoot phenotypes across developmental stages.** The effect of exogenous auxin treatments on mean root length shifts from negative to positive as plants develop across early seedling (A), late seedling (B), and mature stages (C). Plants treated with auxin exhibited higher LR density in the early seedling stage (D) and lower aboveground mass in the late seedling state (E). A significant treatment and genotype and genotype effect, as well as an interaction, was observed for days to flower in plants grown to maturity (F). Statistical significance of treatment (T) and genotype (G) terms are presented for both treatments where $p < 0.001 = $ ***, $p < 0.01 = $ **, and $p > 0.05 = $ ns (Tables S2, S4 and S5). Genotypes are noted by their mutant genes on the right, colored based on gene family (Auxin Binding Protein, Auxin Signaling F Box Protein, Auxin Response Factor, Indole-3-Acetic Acid Inducible, Like Auxin Resistant, PIN-Formed, and SuperRoot) and distinguished by shape with more information on individual genes in Table S1.

when at reproductive maturity. The genotype rank order of root length was distinct in the two treatments (Fig. 5), indicating different mutant lines varied in performance in the control and auxin addition treatment from each other and control lines. For example, while the *pin8* mutant produced longer roots relative to other genotypes in the control treatment at all stages, the line only produced relatively long roots in the early seedling stage when treated with auxin. In plants that were not treated with auxin, we detected significant developmental stage by genotype interactions for belowground mass ($F = 1.8$, $p = 0.0045$; Fig. 5) and aboveground mass ($F = 1.6$, $p = 0.0078$) but not root length ($F = 1.1$, $p = 0.3$). A significant developmental stage by genotype interaction was only observed for

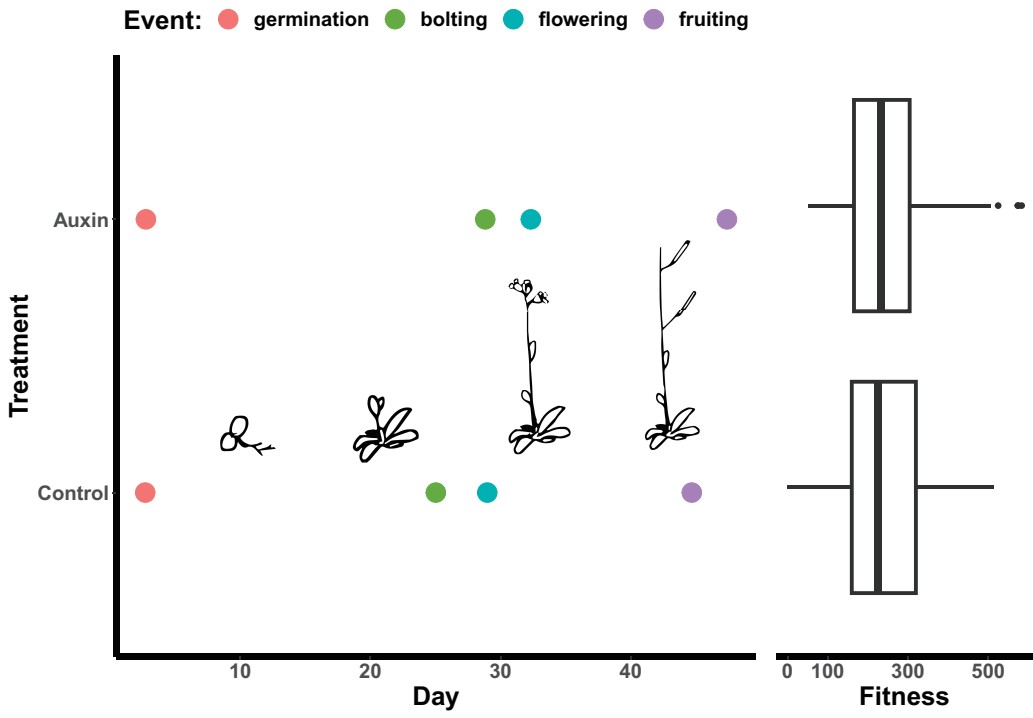

**Figure 2 Effect of auxin treatments on plant phenology.** Auxin treatments had a significant effect on plant phenology by delaying bolting ($F = 24.6$, $p = 2.0e\text{-}06$), subsequent flowering ($F = 16.3$, $p = 9.2e\text{-}05$), and fruiting ($F = 11.1$, $p = 0.0011$). On average, plants treated with auxin entered the reproductive stage (represented by bolting) 3 days later than control plants. This lengthened period of vegetative growth, however, had no effect on plant fitness represented here by the average fruit length in millimeters multiplied by total fruit number.                

belowground mass when plants were treated with auxin ($F = 1.6$, $p = 0.012$), indicating hormone treatment specific phenotypic effects by genotype and developmental stage.

## Variation among APA mutant groups across development

In the control treatment, we did not detect a statistically significant effect of APA group on root length ($F = 0.04$, $p = 0.96$; Fig. 6 and Table S9), belowground mass ($F = 0.06$, $p = 0.94$; Table S9), or aboveground mass in mutants ($F = 0.23$, $p = 0.79$; Table S9). For mutants treated with auxin, a significant APA group effect was observed for root length ($F = 3.3$, $p = 0.04$; Table S9 and Fig. 6), aboveground mass ($F = 4.9$, $p = 0.0082$; Table S9), and belowground mass ($F = 7.1$, $p = 0.0010$; Table S9), with those with 1 or 2 APA sites in mutant genes having the greatest growth. We did not detect any significant developmental stage by APA group interactions for any traits in the control treatment or auxin treatment models (Table S9), such that consistent differences were observed throughout the life cycle. In the auxin treatment, when focusing in on the seedling stage, a significant difference was found between APA groups in early seedling with the 0, and 1 + 2 APA groups having greater aboveground mass than the >2 APA groups (Tukey HSD $p = 0.0262$; Table S10) as well as belowground mass (Tukey HSD $p = 0.0166$; Table S10).

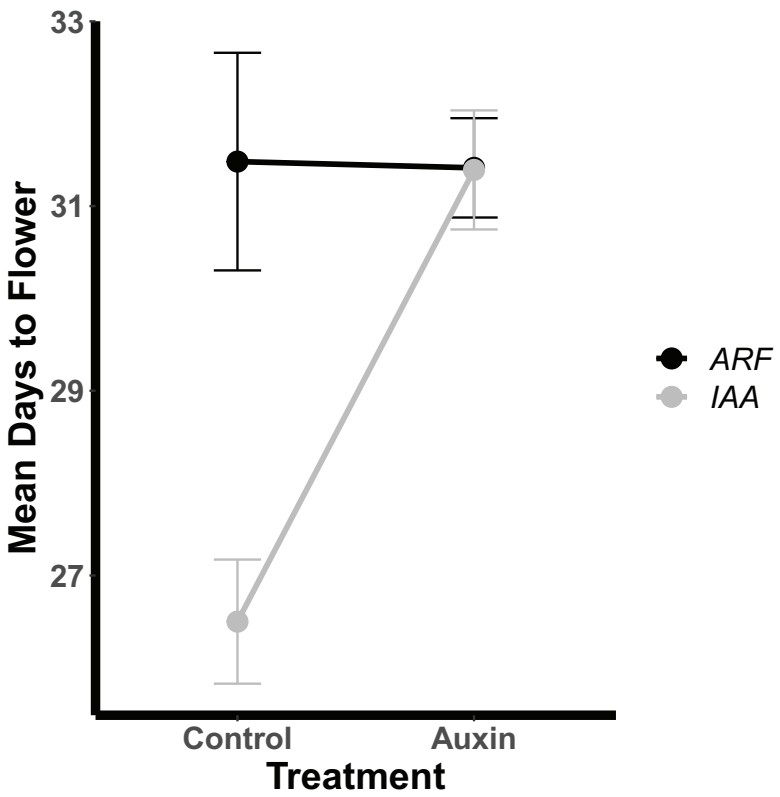

**Figure 3 Mean days to flowering of IAA (Indole-3-Acetic Acid Inducible, black) and ARF (Auxin Response Factor, grey) mutants in control and auxin treatments.** Mean days to flowering of IAA (Indole-3-Acetic Acid Inducible, black) and ARF (Auxin Response Factor, grey) mutants in control and auxin treatments with error bars noting one standard error.

### Variation of natural accession and mutant phenotypes

The variation in root length, belowground mass, and aboveground mass of natural accessions did not differ significantly from that of the insert mutants in the control treatment at any developmental stage (Fig. S5). In the auxin treatment, variation in root length and aboveground mass did not differ at any developmental stage, but natural accessions exhibited significantly greater variation in belowground mass at the mature adult stage when compared to insert mutants (Levene's Test, df = 1, F-value = 3.98, $p = 0.0481$; Fig. S5).

### DISCUSSION

Auxin treatments had a significant positive cumulative effect on aboveground and belowground growth in mature plants and delayed the timing of life stage transitions to reproduction. As we predicted, we found the effect of auxin treatments to be distinct at each developmental stage and to vary across genetic lines, with pronounced differences in root traits across development. We found auxin treatments to have the largest effects on roots traits in the early seedling and mature stages. Seedlings treated with auxin exhibited significantly shorter roots, but when seedlings reached maturity, adult plants treated

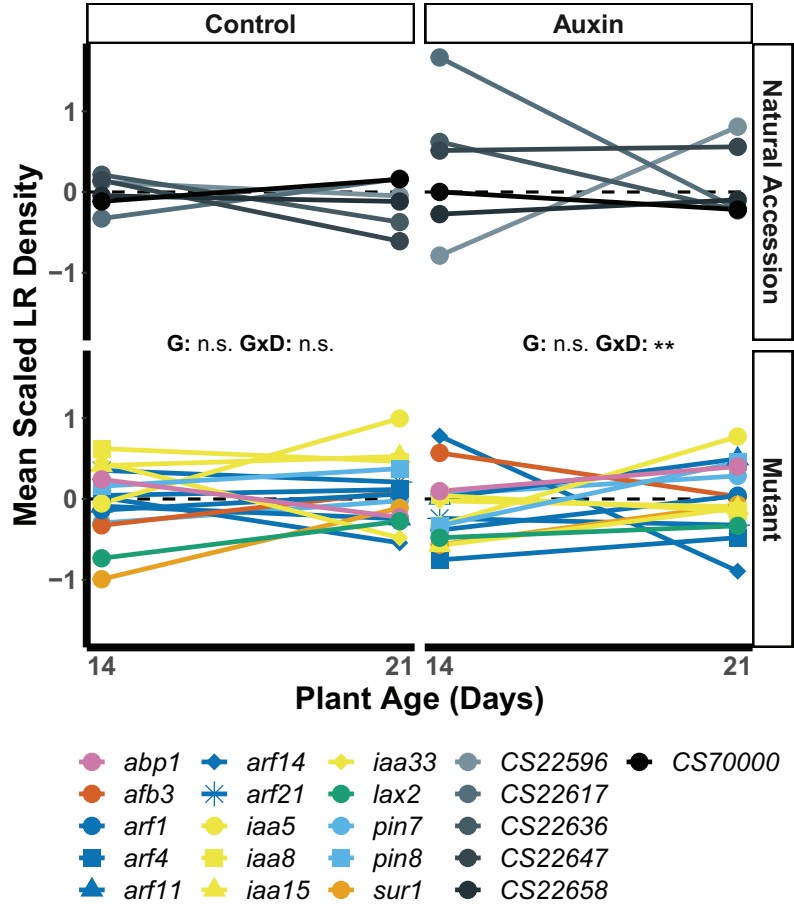

**Figure 4  Mean LR density scaled by plant age and treatment of natural accessions (solid lines) and insert mutants (dashed lines) at 14 and 21 days in the control treatment and when treated with auxin.** After scaling, a significant developmental stage and genotype interaction was observed in the auxin treatment (Table S8). Statistical significance of genotype (G) and genotype by developmental stage interactions (G × D) terms are presented for both treatments where $p < 0.001 = $ ***, $p < 0.01 = $ **, and $p > 0.05 = $ ns (Table S7). Genotypes are noted by their mutant gene or natural accession name below, colored based on mutant gene family (Auxin Binding Protein, Auxin Signaling F Box Protein, Auxin Response Factor, Indole-3-Acetic Acid Inducible, Like Auxin Resistant, PIN-Formed, and SuperRoot; Table S1) and distinguished by shape. Natural accessions are in greyscale and distinguished by shape.

throughout their lifespan with auxin exhibited significantly greater root length. While auxin treatments generally delayed the onset of the reproductive phase, this was not true of all mutant lines. Mutants with inserts in Indole-3-Acetic Acid Inducible genes exhibited later flowering when treated with auxin while those with inserts in Auxin Response Factor genes exhibited earlier flowering or no change in flowering time. Early-stage root phenotypes were not indicative of later stage phenotypes as genotypes shifted in rank order across developmental stages for biomass and root traits. Regardless of differences in whole plant phenotypes across plant development in both treatments, key measures of plant fitness did not differ across treatments or genotypes.

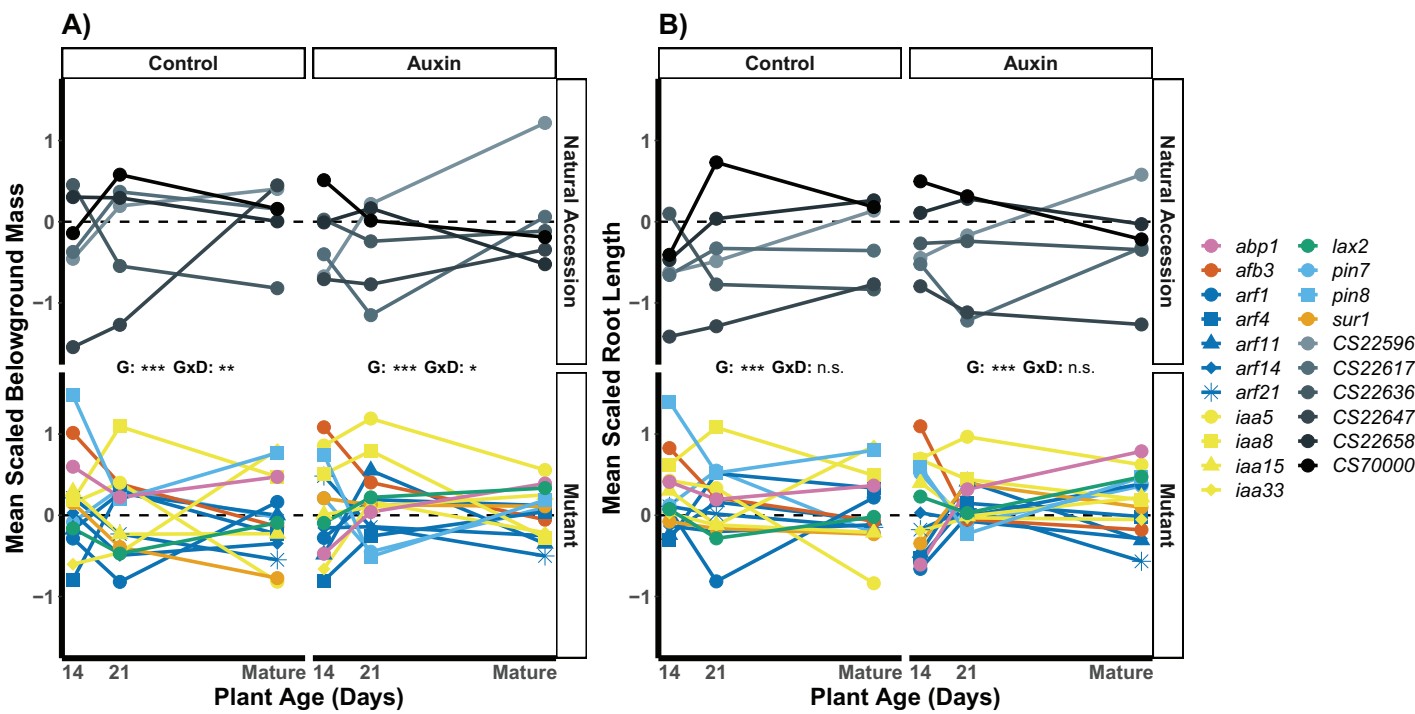

**Figure 5 Mean belowground mass (A) and mean root length (B) scaled by plant age and treatment for natural accessions and insert mutants.** Measurements were recorded at 14, 21 days, and at maturity, which was reached in an average of 44 days in the control treatment and 47 days when plants were treated with auxin. A significant genotype by developmental stage interactions was observed for belowground mass in both treatments but not for root length. Statistical significance of genotype (G) and genotype by developmental stage interactions (G × D) terms are presented for both traits and treatments where $p < 0.001 = ***$, $p < 0.01 = **$, $p < 0.05 = *$, and $p > 0.05 = $ ns (Table S8). Genotypes are noted by their mutant gene or natural accession name on the right, colored based on gene family (Auxin Binding Protein, Auxin Signaling F Box Protein, Auxin Response Factor, Indole-3-Acetic Acid Inducible, Like Auxin Resistant, PIN-Formed, and SuperRoot; Table S1). Natural accessions are in greyscale.

Within developmental stages, significant differences between mutant genotypes could, in part, be explained by the number of APA sites within a mutant gene when treated with auxin but not in the control treatment. Mutants with an intermediate amount of APA sites (1 or 2) in their genes exhibited the greatest plant biomass and the longest root length. Taken together, these results suggest that phenotypes of auxin related mutant genotypes vary in their response to supplemental auxin. Also, there was significant variation in phenotypes of mutant lines across developmental stages that can be partly explained by the number of APA sites within genes.

## Mutant response to auxin treatments

Auxin had a significant effect on the timing of life stage transitions as well as significant effects on belowground and aboveground size traits in early seedlings and mature plants. For example, auxin treatments had a significant negative effect on root length in the early seedling stage, no significant effect in the late seedling stage, and a significant positive effect on root length in the mature adult stage. This suggests that follow up studies on the regulation of *endogenous* auxin based on developmental stage and endogenous auxin production are worthwhile. Focusing on two auxin transport mutants *pin7* and *pin8* and

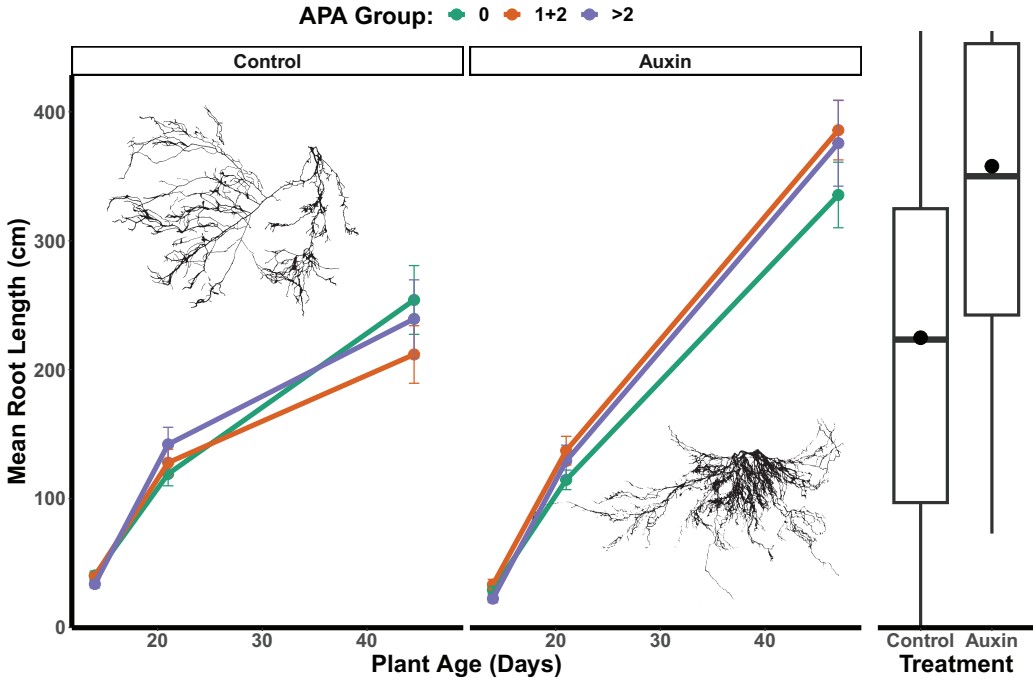

**Figure 6 Mean root length of APA groups across development and auxin treatments.** Mutant lines of genes with 1 or 2 APA sites display significantly greater root length when treated with auxin ($F = 3.3$, $p = 0.040$; Table S9). The box plots to the right indicate overall longer root length maturity in the auxin treatment; means are indicated by circles with medians noted by lines, lower and upper quartile by boxes, and minimums and maximums by whiskers. Example root scans showing typical root highly branched root structures in the auxin treatment in comparison to control.

two signal transduction mutants *arf21* and *iaa8*, both pairs of mutants exhibited a negative response to auxin that carried over from the 14 to the 21 day seedling stage. This shared pattern suggests that these auxin transport and signal transduction genes may have a role in the transition from auxin's negative to positive effect on root growth across development. As shown in our study, auxin applied exogenously does not always inhibit root growth and in fact varies in effect at different stages of development with the contributions of various auxin related genes. We argue that the whole life cycle approach uncovers the cumulative effects of auxin treatments and gene mutations on plant phenotypes and further studies are needed to determine factors leading to potential differences in plant fitness across plant families.

Furthermore, observing transitions between developmental life stages through reproductive maturity adds to our determination of the effect of auxin and auxin related genes on plant phenology. The timing of life stage changes differed significantly by genotype with a significant genotype by treatment interaction being observed for flowering time. Most mutant lines studied here exhibited a later flowering time when treated with auxin, but *sur1*, *arf21*, and *arf4* exhibited an earlier flowering time. Auxin response factor (ARF) mutants exhibit delayed flowering regardless of auxin treatments. ARF proteins repress the expression of auxin related genes when binding to IAA proteins that are

removed by TIR1 in the presence of auxin to allow gene expression (*Petrásek & Friml, 2009*). Previous studies of ARFs in diverse plant species indicate their importance in regulating flower development (*Liu et al., 2015*; *Dong et al., 2022*), and studies of *A. thaliana* ARF mutants report that in the absence of hormone treatments, genotypes with mutant ARF2 genes exhibited delayed flowering (*Okushima et al., 2005*). Mutant ARF proteins may lose their function to repress (or activate) auxin related gene expression in the absence of auxin due to their inability to bind with IAA proteins. But additional research is needed at the molecular scale to determine the mechanism by which mutant ARFs delay flowering time. In contrast, other studies observing delayed flowering in Japanese Morning Glory *Pharbitis nil* following auxin treatments suggest genes outside the auxin related gene families examined here may be contribute to this auxin mediated delay of flowering (*Frankowski et al., 2009*; *Wilmowicz et al., 2014*). Regardless of which specific genes may be involved in this response, auxin contributes to natural variation in the timing of life stage transitions. However, the inclusion of multiple concentrations of exogenous auxin applications to a developmental assay would be a next important step beyond this study to provide more detail on how auxin influences plant phenotypes through a plant's life.

## Variation of root traits across development

Our data suggest that the presence of auxin alters the production of lateral roots in ways that are dependent on plant genotype and age or days of exposure. The seedling phenotypes of all plant genotypes, including insert mutants and natural accessions, differed significantly in control and auxin treatments. LR density, however, was the only seedling trait to exhibit a significant interaction between developmental stage and genotype in the auxin treatment. While a few genotypes exhibited relatively high LR density at the early seedling stage when treated with auxin, they also exhibited relatively low LR density at the late seedling stage. Specifically, the response displayed by the *arf14* mutant LR phenotype suggests ARF14 may play a developmental stage specific role in auxin mediated LR growth and is worthy of further investigation. A significant development by genotype interaction was observed for belowground mass in both treatments indicating developmental tradeoffs. Some plants had greater than average belowground growth in early stages of development that did not persist into later developmental stages. Regardless of these differences in early or late investment in root growth, genotypes did not differ in key measures of plant fitness.

While exogenous auxin was applied to the potting media, changes in the natural production of auxin by belowground and aboveground tissues influence the effect of applications. Just as belowground auxin treatments produced increased rosette diameter when plants bolted in our experiment, auxin produced aboveground contributing to differences in belowground phenotypes across development. Future experiments investigating the levels of endogenous auxin produced at various life stages would be next steps to determine the mechanisms of how auxin production across plant organs alters plant morphology and phenology.

## Variation among APA groups

The number of APA sites within the genes we studied explained differences in plant growth traits when plants were treated with auxin and in early development. As predicted, mutants with inserts in genes without APA sites experiencing less opportunities for variation in regulation across development exhibited decreased root growth. But mutants with inserts in many APA did not display the greatest root lengths. In the early seedling stage, the aboveground and belowground mass of genotypes with mutated genes exhibiting one or two APA sites was significantly higher than that of insert mutants with mutated genes with teo or more APA sites when plants were treated with auxin. Auxin is known to alter the post transcriptional modification of genes *via* poly(A) site switches (*Hong et al., 2018*; *Zeng et al., 2019*). The difference between these APA groups in the auxin treatment may be a result of auxin mediated APA site switches that may have a more drastic effect on plant phenotype when a greater number of APA sites are available. But significant differences between APA groups did not persist in later developmental stages. This suggests APA site number should be considered based on developmental stage when determining candidate genes: something particularly noteworthy for the genetic modification of crops. Differences across APA groups also may be because other regulators contribute to variation to later stage phenotypes such as miRNAs (*Singh et al., 2020*; *Xu et al., 2017*). To fill these gaps, we advocate for additional efforts to draw links between internal molecular phenotypes and the larger scale macroscopic phenotyping (*Großkinsky et al., 2015*), while considering developmental timing for the gene to root phenotypic connections. While our study focused on a specific set of auxin related mutants, there are opportunities to expand this study to other groups of genes in the auxin network, to expose additional genetic function to hormone influenced phenotypes including reproductive fitness. Such expanded studies would inform the influence of the auxin pathway on whole plant phenotypes. Understanding the contributions of gene functions and molecular mechanisms to complex traits expression will aid researchers in developing crops for variable and stressful environments.

## Natural accessions *versus* insert mutants

The variation in observed root length, belowground mass, and aboveground mass across insert mutants is comparable to that of natural accessions. This suggests that inserts in auxin related genes may produce phenotypic variation equivalent to natural genetic variation within our experiment. We advocate for future studies on accessions from diverse environments across several environmental clines to extend our findings and further the evolutionary context of the phenotypic response to hormone variation found here. For example, phenotypic variation among genotypes may only be uncovered in certain natural environments that may provide context for the evolutionary and ecological importance of specific genes (*Tonsor, Alonso-Blanco & Koornneef, 2005*).

Developmental life stage transitions varied between insert mutants and natural accessions. When ARF mutants were in the control treatment they exhibited delayed flowering equivalent to that of other genotypes in the presence of exogenous auxin. The data reveal the role of ARF genes in contributing to annual plant phenology.

Our selection of spring flowering natural accessions was intentional such that the spring life histories would match the experimental timing of the SALK T-DNA mutants from the Col-0 background. This set of lines grown in a stable controlled environment of our experiment. Life cycle variation in *A. thaliana* has previously been shown to be complex with both genetic and environmental factors contributing to the variation in *A. thaliana* life cycles (*Burghardt et al., 2015*). For example, day length and vernalization temperature contribute to variation in *A. thaliana* flowering time (*Reeves & Coupland, 2000*). From our work and these previous studies, a next logical step is to extend these studies of auxin related genes across these ecologically relevant abiotic environments to identify the contribute to differences in *A. thaliana* life cycles.

## CONCLUSIONS

Our investigation of auxin related genes and their role in root development across the entire lifecycle of an annual plant uncovered substantial variation in above and below ground phenotypes that occurs between plant genotypes across development. However, we found the relationships between phenotypic responses and the number of APA sites in mutant genes to be variable with those with fewer APA sites had longer roots but only at early seedling stages. While we found significant differences between whole plant traits across development, the cumulative effects of plant genotype with the exogenous auxin treatment did not result in significant differences in key measures of plant fitness (*e.g.*, fruit number and inflorescence branching), despite other phenotypic differences at distinct life stages. Additionally, the variation in how mutants from distinct gene families (ARF and IAA) respond to auxin treatments highlights the key regulation of specific whole plant morphology and phenology in an auxin dependent manner. The root system differences of genotypes with mutations in genes exhibiting varying APA when treated with auxin suggest that auxin not only regulates the expression of genes, *but also their post-transcription modification*. We generally observed mutants with inserts in genes with fewer APA sites to exhibit higher biomass when treated with auxin suggesting genes with a greater number of APA sites and their regulation *via* auxin to be more crucial to plant growth across development. While our studies of exogenous auxin application uncovered key contributions of development, gene family and APA number on root and shoot phenotypic expression, we advocate future studies to determine how the production of hormones *de novo* varies across development and connect to reproductive fitness measures.

Understanding the development of root systems and the role of roots in nutrient uptake, anchoring, and nutrient storage within model plant systems can provide the next logical step to examining the root systems of domesticated cereal crops and wild species (*Smith & De Smet, 2012*; *Hodge, 2006*; *De Baets et al., 2006*; *Hales et al., 2009*). Gene action variation across development and in response to auxin may optimize the structure and function of root systems as plant nutrient demands and soil nutrient availability fluctuate across the lifecycle of a plant. Root systems are shaped by natural selection through internal biotic processes including auxin synthesis, transport, and signaling, as well as gene regulation *via* polyadenylation, result in variation across genotypes and life stages, as we demonstrate

here. Further developmental studies will extend the context of plant hormones in the evolutionary ecology of natural populations and to inform crop breeding programs.

## ACKNOWLEDGEMENTS

We thank April Bisner, unPAK project manager, and members of the Murren and Rutter labs for assistance during the project.

### Funding

This work was supported by USDA-REEU, the College of Charleston School of Science Mathematics and Engineering and NSF IOS 1052262. There was no additional external funding received for this study. The funders had no role in study design, data collection and analysis, decision to publish, or preparation of the manuscript.

### Grant Disclosures

The following grant information was disclosed by the authors:
USDA-REEU, the College of Charleston School of Science Mathematics and Engineering and NSF IOS 1052262.

### Competing Interests

The authors declare that they have no competing interests.

### Author Contributions

- Patrick Sydow conceived and designed the experiments, performed the experiments, analyzed the data, prepared figures and/or tables, authored or reviewed drafts of the article, and approved the final draft.
- Courtney J. Murren conceived and designed the experiments, analyzed the data, authored or reviewed drafts of the article, and approved the final draft.

### Data Availability

The data are available in the Supplemental Files and at GitHub: https://GitHub.com/sydowpw/SydowMurrenPeerJ2024.

### Supplemental Information

Supplemental information for this article can be found online at http://dx.doi.org/10.7717/peerj.16873#supplemental-information.

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
