# Peer review of "Above and belowground phenotypic response to exogenous auxin across Arabidopsis thaliana mutants and natural accessions varies from seedling to reproductive maturity"

_PeerJ, doi:10.7717/peerj.16873_

## Round 0.1 · original submission · Major Revisions

I recommend major revisions to the manuscript, focusing on title specificity, updated citations, and a clearer hypothesis. Reviewers 1 and 3 provided important detailed remarks. Please, ensure clarity and objectivity in materials, procedures, and results, while consolidating figures and tables for better presentation. Incorporate recent literature and action mechanisms in the discussion, and provide results-based statements in the conclusion. Please, consider multiple IAA concentrations for a comprehensive analysis.

Reviewer 1 ·

Basic reporting

The title is too general and does not specify which types of mutants were examined.

The introduction is adequate but should include new citations on the subject of alternative polyadenylation (APA) and focus more on this description than on the already-known effects of auxin.

Materials and procedures should be rewritten, streamlined, and made more objective. I found it confusing, and with sections that are a bibliography review, such as the Study Species topic (an entirely unnecessary topic for the Arabidopsis, which is well-known to all). I'm unsure if the topic Synthesis of Literature is necessary given that it was simply a method for evaluating the best days of analysis, and these results are supplementary data. This subject could be under the "Treatments." topic. The Measurements section should be simplified and made more objective in order to be rewritten and condensed. The subject of statistical analysis is vast.

Results The topic Synthesis of Literature does not need to be presented separately and must be included as a supplement to the other results. It doesn't seem like the format, which divides topics by plant age, is the right way to start a paragraph in a scientific paper. This format reminds me more of a report. I recommend rewriting in a more formal and objective style; the results are extremely tiresome to read.

Since they are statistical analyses of the presented graphs, the tables are so repetitive that they could be combined, better explained, and placed as a supplement.

The figure captions are inadequate, as there is no description of the genotypes (the reader must remember to consult another supplementary table in the article). Where can I find WT plants in Grafis?

The vast quantity of mutants and strains makes visualization challenging. It is not presented well, so I suggest removing some or attempting a different method so that we can distinguish between each item. For instance, to separate by mutant group (separating the lineages).
The figures and their captions need improvement.

This article contains numerous figures that can be grouped together (Figures 1 and 2, for instance). The caption for Figure 4 is inadequate, and the figure itself is of poor quality. This phenological analysis is difficult to comprehend and is not included in the materials and methods. I did not comprehend the section on fitness. It is unclear how this figure should be interpreted.
The figures are so similar and repetitive that they appear identical to me. (as you should combine Figures 6 and 7 and 8 into a single figure with the letters A B C D E F.

Figure 9 is unclear, and the depiction of the roots makes it even more so. I was unable to comprehend the purpose of this figure.
The discussion requires improvement. The quotations are very old and do not explain the possible action mechanisms in each case. If there are no results in this paper that demonstrate this hormonal interaction, there is no reason to discuss other hormones such as ethylene and ABA in this manner.

It is necessary to revise the discussion so that it incorporates the results obtained from recent literature and can correlate the modes of action and mutant responses. The way it is separated (subtopics) makes it appear as though they are distinct, but they are not. I suggest reviewing the manner in which the data is discussed.
The correlation of their functions with the obtained responses (mutants) and number of APA sites may be a viable option.

Experimental design

The experimental design seems adequate for what is proposed.

Validity of the findings

The article does not show any innovative results or that can contribute significantly.
The statistical analyzes are robust but the analyzes themselves do not add much to what the article proposes.

Reviewer 2 ·

Basic reporting

In general, the manuscript is well written and structured, with a commendable amount of detail especially in the methods section, which will ensure a high degree of reproducibility. Statistical analysis is not my area of expertise, but in general the data appear to be logically and well presented, and support the authors’ conclusions. I would suggest a few small changes to the manuscript detailed as follows:
1. The authors repeatedly describe ARFs as repressing gene expression, whereas, ARFs can be repressing (most of them) or activating (5 ARFs). The text should reflect this activating and repressing activity of ARFs.
2. While the authors propose to have investigated the role of auxin in regulating polyadenylation in auxin related genes, and how this affects their activity by using T-DNA insertion mutants, in practice, it is not fully clear how these mutants were selected, and further the effect of the mutation on polyadenylation. T-DNA insertions have a range of effects on gene expression, resulting in complete loss-of- expression to downregulation, but not all of these would be linked to changes in PA. It would be nice to see a bit more of a link between PA and these insertion mutants. What effects do these insertions have on PA? How can we be sure that the differences in phenotypes between WT and auxin mutants are because of changes in PA? Ideally it would be nice to see the effect of the T-DNA insertion on polyadenylation, but I am not even sure if this is possible, and I would certainly not expect the authors to do this for this manuscript. However, at least a schematic figure showing the position of the PA sites within the gene sequence and how the T-DNA insertion affects these would be helpful.

Experimental design

The experiments are well designed, with plenty of background from the literature provided supporting hypotheses and experimental design. Data is presented as a sufficient number of biological replicates. The data is presented to a high standard and the experiments are logical.

Validity of the findings

In this study, the authors have investigated how auxin-dependent polyadenylation in auxin signalling related genes influences different parameters of plant growth in both, early and late stages of plant development, using the model plant Arabidopsis thaliana.

The authors emphasize that their study fulfills a crucial knowledge gap in the plant science community – the role of auxin in the regulation of plant development during the latter stages of the Arabidopsis life cycle. In this regard, they are absolutely correct – most studies investigating the role of auxin in root development (as an example) have focused on young plants that are typically 10-14 days old. There are some very interesting and novel findings reported here, such as for example, that auxin seems to have different effects on root length in young and older plants. I have no concerns for this section.

Reviewer 3 ·

Basic reporting

I have read the manuscript entitled “Root variation in response to auxin across genotypes: from seedling to maturity” submitted to Peer J.

In short, the authors used a range of genotypes of the Arabidopsis that change auxin transport, synthesis, metabolism, or signal transduction to analyse the growth pattern in response to exogenous auxin. The data are robust and bring many specific results considering the specific changes observed in the genotypes. However, many issues can be raised considering the scientific background, material and methods, discussion, and conclusion.

Experimental design

Scientific background:
1. The citations should be more recent.
2. The authors raised some questions (Line 126) based on genotypes and exogenous AA, but the hypothesis is not clear. The introduction has much didactic information about auxin, but it should be associated with an evident hipothesis.

Matherial and methods:
1. Line 155: The botanical information about the Arabidopsis species is outside the scope of the manuscript.
2. The authors used only the concentration of IAA. What references the authors have based on? It is not surprised that exogenous hormones have many distinct signaling pathways if applied at different concentrations. The use of only one concentration of IAA makes the discussion of the results hard and makes the conclusion inconsistent. Very low, low, and high doses of IAA would be suitable considering the dose-response curve.

Validity of the findings

Discussion:
1. The discussion of results should be followed by a number of figures or tables.
2. Line 442 to 445: From what work the sentence is?
3. Line 447 to 450: Many interactions can occur between auxin and other hormones. The discussion with ethylene is speculative beyond the results.
4. Line 456: What means "Additional auxin"? Is an exogenous application? If so, in what figure or table can the results be observed?
5. Line 471 to 475: The discussion is speculative because the authors compared their results with those of other species and the use of ethylene.

Additional comments

Conclusions
1. There are no conclusions from the results, but just perspectives.

---

## Round 0.2 · Minor Revisions

The reviewer commends the improved clarity and streamlined presentation in this revision. Your revised manuscript requires a few minor clarifications on root traits, gene mutant phenotypes, plant age-related observations, and 'pin8' terminology. Please also correct figure legends for data representation and color coding.

Reviewer 2 ·

Basic reporting

This article is a revision of a previously reviewed version. Overall, this version is much more streamlined, and the findings are reported more concisely, which makes overall readability much easier. I did not have any major concerns with the previous version, and I find this version vastly improved. There are some minor revisions detailed as below:

1. Line 152: What does ‘root traits be greater’ mean? Does this refer to root biomass, rooting density, root length, or all of these? The authors should be more specific here.
2. Line 160: I don’t really understand what the authors mean here. Why would mutants in genes with fewer polyA sites be expected to have fewer lateral roots? Given that these are insertion mutants, they are loss-of-function mutants (theoretically) and this hypothesis would only hold true if all the genes tested are known to be positive regulators of lateral rooting. Is this the case? If so, the authors should specify this somewhere.
3. Lines 326-328: Do the authors mean that they observed smaller rosettes and decreased biomass in both older and younger plants? This is not clear, since the term ‘In contrast’ has been used. Please clarify
4. Line 373: What is pin8? Do you mean the WT version of the gene, since you refer to ‘the mutant’ in the same sentence, which I assume is the insertion mutant?

Figures:
Fig 2: Is this cumulative data from all lines? Or which specific lines? This is not specified in the legend either, please elaborate.

Fig3: The legend says that ARF genes are denoted in grey, but they are denoted as black in the figure. Please clarify and fix this.

Experimental design

The experimental design seems logical for the proposed hypotheses, and this section of text has been vastly improved by being more concise. I will strongly suggest that that authors annotate the WT controls in the graphs, wherever these are present to improve readability.

Validity of the findings

Broadly, the experimental techniques and statistical analyses are robust. I do not have any suggested changes here.

---

## Round 0.3 · accepted · Accept

All comments from the previous reviewers have been adequately addressed. I have personally assessed the revisions and am satisfied with the improvements made to the study. The manuscript provides insightful findings on how auxin-related mutants can have distinct root and shoot phenotypic expression in Arabidopsis thaliana, revealing significant differences in response to auxin treatments across different life stages and genotypes. The study's contribution to understanding the interactions between genotype, developmental stage, and hormone treatment in plant traits and fitness is noteworthy. I believe this manuscript is now ready for publication.